# Characteristic Circuits

**Zhongjie Yu**
TU Darmstadt
Darmstadt, Germany

**Martin Trapp**
Aalto University
Espoo, Finland

**Kristian Kersting**
TU Darmstadt/Hessian.AI/DFKI
Darmstadt, Germany

{yu,kersting}@cs.tu-darmstadt.de, martin.trapp@aalto.fi

## Abstract

In many real-world scenarios, it is crucial to be able to *reliably* and *efficiently* reason under uncertainty while capturing complex relationships in data. Probabilistic circuits (PCs), a prominent family of *tractable* probabilistic models, offer a remedy to this challenge by composing simple, tractable distributions into a high-dimensional probability distribution. However, learning PCs on heterogeneous data is challenging and densities of some parametric distributions are not available in closed form, limiting their potential use. We introduce characteristic circuits (CCs), a family of tractable probabilistic models providing a unified formalization of distributions over heterogeneous data in the spectral domain. The one-to-one relationship between characteristic functions and probability measures enables us to learn high-dimensional distributions on heterogeneous data domains and facilitates efficient probabilistic inference even when no closed-form density function is available. We show that the structure and parameters of CCs can be learned efficiently from the data and find that CCs outperform state-of-the-art density estimators for heterogeneous data domains on common benchmark data sets.

## 1 Introduction

Probabilistic circuits (PCs [Choi et al., 2020]) have gained increasing attention in the machine learning community as a promising modelling family that renders many probabilistic inferences tractable with little compromise in their expressivity. Their beneficial properties have prompted many successful applications in density estimation [*e.g.*, Peharz et al., 2020, Di Mauro et al., 2021, Dang et al., 2022, Correia et al., 2023] and in areas where probabilistic reasoning is key, for example, neuro-symbolic reasoning [Ahmed et al., 2022], certified fairness [Selvam et al., 2023], or causality [Zečević et al., 2021]. Moreover, recent works have explored ways of specifying PCs for more complex modelling scenarios, such as time series [Trapp et al., 2020, Yu et al., 2021b,a] or tractable representation of graphs [Wang et al., 2022].

Figure 1: Characteristic circuits provide a unified, tractable specification of joint continuous and discrete distributions in the spectral domain of probability measures.

However, while density estimation is at the very core of many machine learning techniques (*e.g.*, approximate Bayesian inference [Murphy, 2012]) and a fundamental tool in statistics to identify characteristics of the data such as $k^{\text{th}}$ order moments or multimodality [Silverman, 2018], even in the case of parametric families, densities are sometimes

not available in closed-form. For example, only special cases of $\alpha$-stable distributions provide closed-form densities [Nolan, 2013]. Fortunately, there exists a one-to-one correspondence between probability measures and characteristic functions [Sasvári, 2013], which can be understood as the Fourier-Stieltjes transform of the probability measures, enabling the characterisation of any probability measure through its characteristic function. Henceforth, the characteristic function of probability measures has found wide applicability in statistics, ranging from its use as a non-parametric estimator through the empirical characteristic function [Feuerverger and Mureika, 1977] to estimate heavy-tailed data, *e.g.*, through the family of $\alpha$-stable distributions [Nolan, 2013]. However, even though the characteristic function has many beneficial properties, its application to encode high-dimensional data distributions and efficient computation of densities can be quite challenging [Nolan, 2013].

In this work, we bridge between the characteristic function of probability measures and PCs. We do so by examining PCs from a more general perspective, similar in spirit to their specifications as a summation over functions on a commutative semiring [Friesen and Domingos, 2016] or as a convex combination of product measures on product probability spaces [Trapp et al., 2020]. Instead of defining the circuit over density functions, we propose to form the circuit over the *characteristic function* of the respective probability measures, illustrated in Fig. 1. The resulting *characteristic circuits* (CCs) are related to recent works, which define a circuit over probability generating polynomials to represent discrete probability distributions [Zhang et al., 2021], in that both approaches can be understood as transformation methods. The benefits of using the spectral domain are manifold: (i) *characteristic functions* as the base enable a *unified view* for discrete and continuous random variables, (ii) directly representing the *characteristic function* allows learning distributions that *do not* have closed-form expressions for their density, and (iii) the moment can be obtained efficiently by differentiating the circuit. When modelling heterogeneous data, standard PCs do not naturally lend themselves to a unified view of mixed data but treat discrete and continuous random variables (RVs) conceptually differently. The difference arises as PCs model heterogeneous data domains *after* integration w.r.t. the base measure which, in the case of mixed domains, differs between discrete and continuous RVs. RVs distributed according to a singular distribution can typically not be represented in PCs at all. This dependence on the base measure is subtly embedded within PCs, resulting in challenges when it comes to learning these models in heterogeneous domains. In contrast, CCs provide a unified view compared to PCs by moving away from the dependence on the base measure, achieved by representing the distribution through its characteristic function, which is independent of the base measure.

In summary, our contributions are: (1) We propose characteristic circuits, a novel deep probabilistic model class representing the joint of discrete and continuous random variables through a unifying view in the spectral domain. (2) We show that characteristic circuits retain the tractability of PCs despite the change of domain and enable efficient computation of densities, marginals, and conditionals. (3) We derive parameter and structure learning for characteristic circuits and find that characteristic circuits outperform SOTA density estimators in the majority of tested benchmarks.[1]

We proceed as follows. We start by discussing related work and review preliminaries on probabilistic circuits and characteristic functions. Consequently, we define our model *characteristic circuits*, discuss theoretical properties, and show how to learn the circuits' parameters and structure. We conclude by presenting an empirical evaluation and discussion of the new model class.

## 2 Related Work

**Characteristic functions** (CFs) were originally proposed as a tool in the study of limit theorems and afterwards developed with independent mathematical interest [Lukacs, 1972]. The uniqueness between CFs and probability measures is recovered with Lévy's inversion theorem [Sasvári, 2013]. A popular application of the CF is in statistical tests [*e.g.*, Eriksson and Koivunen, 2003, Su and White, 2007, Wang and Hong, 2018, Ke and Yin, 2019]. In practice, the CF of a distribution is in most cases not easy to estimate, and in turn, the empirical characteristic function (ECF) is employed as an approximation to the CF [Feuerverger and Mureika, 1977]. The ECF has been successfully applied in sequential data analysis tasks [Knight and Yu, 2002, Yu, 2004, Davis et al., 2021]. When handling high dimensional data, multivariate CFs, and ECFs were proposed for modelling *e.g.* multivariate time series [Lee et al., 2022] and images [Ansari et al., 2020]. Although a mass of work has been

---

[1]Source code is available at `https://github.com/ml-research/CharacteristicCircuits`

developed for the applications of CF, less attention has been paid to estimating the model quality of CF itself. Therefore, modelling the multivariate CF remains to be challenging.

**Probabilistic circuits** (PCs) are a unifying framework for tractable probabilistic models [Choi et al., 2020] that recently show their power in *e.g.* probabilistic density estimation [Dang et al., 2020, Di Mauro et al., 2021, Zhang et al., 2021], flexible inference [Shao et al., 2022], variational inference [Shih and Ermon, 2020], and sample generating [Peharz et al., 2020]. When it comes to data containing both discrete and continuous values, a mixture of discrete and continuous random variables is employed in PCs. Molina et al. [2018] propose to model mixed data based on the Sum-Product Network (SPN) structure, casting the randomized dependency coefficient [Lopez-Paz et al., 2013] for independence test for hybrid domains and piece-wise polynomial as leaf distributions, resulting in Mixed Sum-Product Networks (MSPN). Furthermore, statistical data type and likelihood discovery have been made available with Automatic Bayesian Density Analysis (ABDA) [Vergari et al., 2019], which is a suitable tool for the analysis of mixed discrete and continuous tabular data. Moreover, Bayesian SPNs [Trapp et al., 2019] use a well-principled Bayesian framework for SPN structure learning, achieving competitive results in density estimation on heterogeneous data sets. The above-mentioned models try to handle the probability measure with either parametric density/mass functions or histograms, but yet could not offer a unified view of heterogeneous data. PCs will also fail to model leaves with distributions that do not have closed-form density expressions.

# 3 Preliminaries on Probabilistic Circuits and Characteristic Functions

Before introducing characteristic circuits, we recap probabilistic circuits and characteristic functions.

## 3.1 Probabilistic Circuits (PCs)

PCs are tractable probabilistic models, structured as rooted directed acyclic graphs, where each *leaf* node L represents a probability distribution over a univariate RV, each *sum* node S models a mixture of its children, and each *product* node P models a product distribution (assuming independence) of their children. A PC over a set of RVs $\boldsymbol{X}$ can be viewed as a computational graph $\mathcal{G}$ representing a tractable probability distribution over $\boldsymbol{X}$, and the value obtained at the root node is the probability computed by the circuit. We refer to Choi et al. [2020] for more details.

Each node in $\mathcal{G}$ is associated with a subset of $\boldsymbol{X}$ called the scope of a node N and is denoted as $\psi(\mathsf{N})$. The scope of an inner node is the union of the scope of its children. Sum nodes compute a weighted sum of their children $\mathsf{S} = \sum_{\mathsf{N} \in \mathrm{ch}(\mathsf{S})} w_{\mathsf{S},\mathsf{N}} \mathsf{N}$, and product nodes compute the product of their children $\mathsf{P} = \prod_{\mathsf{N} \in \mathrm{ch}(\mathsf{P})} \mathsf{N}$, where $\mathrm{ch}(\cdot)$ denotes the children of a node. The weights $w_{\mathsf{S},\mathsf{N}}$ are generally assumed to be non-negative and normalized (sum up to one) at each sum node. We also assume the PC to be smooth (complete) and decomposable [Darwiche, 2003], where smooth requires all children of a sum node having the same scope, and decomposable means all children of a product node having pairwise disjoint scopes.

## 3.2 Characteristic Functions (CFs)

Characteristic functions provide a *unified view* for discrete and continuous random variables through the Fourier–Stieltjes transform of their probability measures. Let $\boldsymbol{X}$ be a $d$-dimensional random vector, the CF of $\boldsymbol{X}$ for $\boldsymbol{t} \in \mathbb{R}^d$ is given as:

$$\varphi_{\boldsymbol{X}}(\boldsymbol{t}) = \mathbb{E}[\exp(\mathrm{i}\,\boldsymbol{t}^\top \boldsymbol{X})] = \int_{\boldsymbol{x} \in \mathbb{R}^d} \exp(\mathrm{i}\,\boldsymbol{t}^\top \boldsymbol{x}) \, \mu_{\boldsymbol{X}}(\mathrm{d}\boldsymbol{x}), \tag{1}$$

where $\mu_{\boldsymbol{X}}$ is the distribution/probability measure of $\boldsymbol{X}$. CFs have certain useful properties. We will briefly review those that are relevant for the remaining discussion: (i) $\varphi_X(0) = 1$ and $|\varphi_X(t)| \leq 1$; (ii) for any two RVs $X_1$, $X_2$, both have the same distribution iff $\varphi_{X_1} = \varphi_{X_2}$; (iii) if $X$ has $k$ moments, then $\varphi_X$ is $k$-times differentiable; and (iv) two RVs $X_1$, $X_2$ are independent iff $\varphi_{X_1, X_2}(s, t) = \varphi_{X_1}(s)\varphi_{X_2}(t)$. We refer to Sasvári [2013] for a more detailed discussion of CFs and their properties.

**Theorem 3.1** (Lévy's inversion theorem [Sasvári, 2013]). *Let $X$ be a real-valued random variable, $\mu_X$ its probability measure, and $\varphi_X \colon \mathbb{R} \to \mathbb{C}$ its characteristic function. Then for any $a, b \in \mathbb{R}$,*

*a < b, we have that*

$$\lim_{T \to \infty} \frac{1}{2\pi} \int_{-T}^{T} \frac{\exp(-\mathrm{i}ta) - \exp(-\mathrm{i}tb)}{\mathrm{i}t} \varphi_X(t)\mathrm{d}t = \mu_X[(a,b)] + \frac{1}{2}(\mu_X(a) + \mu_X(b)) \quad (2)$$

*and, hence, $\varphi_X$ uniquely determines $\mu_X$.*

**Corollary.** *If $\int_{\mathbb{R}} |\varphi_X(t)|\mathrm{d}t < \infty$, then $X$ has a continuous probability density function $f_x$ given by*

$$f_X(x) = \frac{1}{2\pi} \int_{\mathbb{R}} \exp(-\mathrm{i}tx)\varphi_X(t)\mathrm{d}t. \quad (3)$$

Note that not every probability measure admits an analytical solution to Eq. (3), *e.g.*, only special cases of the family of $\alpha$-stable distributions have a closed-form density function [Nolan, 2013], and numerical integration might be needed.

**Empirical Characteristic Function (ECF).** In many cases, a parametric form of the data distribution is not available and one needs to use a non-parametric estimator. The ECF [Feuerverger and Mureika, 1977, Cramér, 1999] is an unbiased and consistent non-parametric estimator of the population characteristic function. Given data $\{\boldsymbol{x}_j\}_{j=1}^n$ the ECF is given by

$$\hat{\varphi}_{\mathbb{P}}(\boldsymbol{t}) = \frac{1}{n} \sum_{j=1}^{n} \exp(\mathrm{i}\,\boldsymbol{t}^\top \boldsymbol{x}_j). \quad (4)$$

**Evaluation Metric.** To measure the distance between two distributions represented by their characteristic functions, the squared characteristic function distance (CFD) can be employed. The CFD between two distributions $\mathbb{P}$ and $\mathbb{Q}$ is defined as:

$$\mathrm{CFD}_\omega^2(\mathbb{P}, \mathbb{Q}) = \int_{\mathbb{R}^d} |\varphi_{\mathbb{P}}(\boldsymbol{t}) - \varphi_{\mathbb{Q}}(\boldsymbol{t})|^2 \, \omega(\boldsymbol{t};\eta)\mathrm{d}\boldsymbol{t}, \quad (5)$$

where $\omega(\boldsymbol{t};\eta)$ is a weighting function parameterized by $\eta$ and guarantees the integral in Eq. (5) converge. When $\omega(\boldsymbol{t};\eta)$ is a probability density function, Eq. (5) can be rewritten as:

$$\mathrm{CFD}_\omega^2(\mathbb{P}, \mathbb{Q}) = \mathbb{E}_{\boldsymbol{t} \sim \omega(\boldsymbol{t};\eta)} \left[ |\varphi_{\mathbb{P}}(\boldsymbol{t}) - \varphi_{\mathbb{Q}}(\boldsymbol{t})|^2 \right]. \quad (6)$$

Actually, using the uniqueness theorem of CFs, we have $\mathrm{CFD}_\omega(\mathbb{P}, \mathbb{Q}) = 0$ iff $\mathbb{P} = \mathbb{Q}$ [Sriperumbudur et al., 2010]. Computing Eq. (6) is generally intractable, therefore, we use Monte-Carlo integration to approximate the expectation, resulting in $\mathrm{CFD}_\omega^2(\mathbb{P}, \mathbb{Q}) \approx \frac{1}{k} \sum_{j=1}^{k} |\varphi_{\mathbb{P}}(t_j) - \varphi_{\mathbb{Q}}(t_j)|^2$, where $\{t_1, \cdots, t_k\} \overset{\text{i.i.d.}}{\sim} \omega(\boldsymbol{t};\eta)$. We refer to Ansari et al. [2020] for a detailed discussion.

# 4  Characteristic Circuits

Now we have everything at hand to introduce characteristic circuits. We first give a recursive definition of CC, followed by devising each type of node in a CC. We then show CCs feature efficient computation of densities, and in the end, introduce how to learn a CC from data.

**Definition 4.1** (Characteristic Circuit). *Let $\boldsymbol{X} = \{X_1, \ldots, X_d\}$ be a set of random variables. A characteristic circuit denoted as $\mathcal{C}$ is a tuple consisting of a rooted directed acyclic graph $\mathcal{G}$, a scope function $\psi\colon \mathrm{V}(\mathcal{G}) \to \mathcal{P}(\boldsymbol{X})$, parameterized by a set of graph parameters $\theta_\mathcal{G}$. Nodes in $\mathcal{G}$ are either sum (S), product (P), or leaf (L) nodes. With this, a characteristic circuit is defined recursively as follows:*

1. *a characteristic function for a scalar random variable is a characteristic circuit.*

2. *a product of characteristic circuits is a characteristic circuit.*

3. *a convex combination of characteristic circuits is a characteristic circuit.*

Let us now provide some more details. To this end, we denote with $\varphi_\mathcal{C}(\boldsymbol{t})$ the output of $\mathcal{C}$ computed at the root of $\mathcal{C}$, which represents the estimation of characteristic function given argument of the characteristic function $\boldsymbol{t} \in \mathbb{R}^d$. Further, we denote the number of RVs in the scope of N as $p_\mathsf{N} :=$

$|\psi(\mathsf{N})|$ and use $\varphi_\mathsf{N}(\boldsymbol{t})$ for the characteristic function of a node. Throughout the paper, we assume the CC to be smooth and decomposable.

**Product Nodes.** A product node in a CC encodes the independence of its children. Let $X$ and $Y$ be two RVs. Following property (iv) of characteristic functions, the characteristic function of $X, Y$ is given as $\varphi_{X,Y}(t, s) = \varphi_X(t)\varphi_Y(s)$, if and only if $X$ and $Y$ are independent. Therefore, by definition, the characteristic function of product nodes is given as:

$$\varphi_\mathsf{P}(\boldsymbol{t}) = \prod\nolimits_{\mathsf{N}\in\mathrm{ch}(\mathsf{P})} \varphi_\mathsf{N}(\boldsymbol{t}_{\psi(\mathsf{N})}), \tag{7}$$

where $\boldsymbol{t} = \bigcup_{\mathsf{N}\in\mathrm{ch}(\mathsf{P})} \boldsymbol{t}_{\psi(\mathsf{N})}$.

**Sum Nodes.** A sum node in a CC encodes the mixture of its children. Let the parameters of S be given as $\sum_{\mathsf{N}\in\mathrm{ch}(\mathsf{S})} w_{\mathsf{S},\mathsf{N}} = 1$ and $w_{\mathsf{S},\mathsf{N}} \geq 0, \forall\mathsf{S}, \mathsf{N}$. Then the sum node in a CC is given as: $\varphi_\mathsf{S}(\boldsymbol{t}) =$

$$\int_{\boldsymbol{x}\in\mathbb{R}^d} \exp(\mathrm{i}\boldsymbol{t}^\top\boldsymbol{x}) \left[ \sum\nolimits_{\mathsf{N}\in\mathrm{ch}(\mathsf{S})} w_{\mathsf{S},\mathsf{N}} \, \mu_\mathsf{N}(\mathrm{d}\boldsymbol{x}) \right] = \sum\nolimits_{\mathsf{N}\in\mathrm{ch}(\mathsf{S})} w_{\mathsf{S},\mathsf{N}} \underbrace{\int_{\boldsymbol{x}\in\mathbb{R}^{p_\mathsf{S}}} \exp(\mathrm{i}\boldsymbol{t}^\top\boldsymbol{x}) \, \mu_\mathsf{N}(\mathrm{d}\boldsymbol{x})}_{=\varphi_\mathsf{N}(\boldsymbol{t})}. \tag{8}$$

**Leaf Nodes.** A leaf node of a CC models the characteristic function of a univariate RV. To handle various data types, we propose the following variants of leaf nodes.

*ECF leaf.* The most straightforward way for modelling the leaf node is to directly employ the empirical characteristic function for the local data at each leaf, defined as $\varphi_{\mathsf{L}_\mathrm{ECF}}(t) = \frac{1}{n}\sum_{j=1}^n \exp(\mathrm{i}\,t\,x_j)$, where $n$ is the number of instances at the leaf L, and $x_j$ is the $j^{th}$ instance. The ECF leaf is non-parametric and is determined by the $n$ instances $x_j$ at the leaf.

*Parametric leaf for continuous RVs.* Motivated by existing SPN literature, we can assume that the RV at a leaf node follows a parametric continuous distribution *e.g.* normal distribution. With this, the leaf node is equipped with the CF of normal distribution $\varphi_{\mathsf{L}_\mathrm{Normal}}(t) = \exp(\mathrm{i}\,t\,\mu - \frac{1}{2}\sigma^2 t^2)$, where parameters $\mu$ and $\sigma^2$ are the mean and variance.

*Parametric leaf for discrete RVs.* For discrete RVs, if it is assumed to follow categorical distribution ($P(X = j) = p_j$), then the CF at the leaf node can be defined as $\varphi_{\mathsf{L}_\mathrm{Categorical}}(t) = \mathbb{E}[\exp(\mathrm{i}\,t\,x)] = \sum_{j=1}^k p_j \exp(\mathrm{i}\,t\,j)$. Other discrete distributions which are widely used in probabilistic circuits can also be employed as leaf nodes in CCs, *e.g.*, Bernoulli, Poisson, and geometric distributions.

*$\alpha$-stable leaf.* In the case of financial data or data distributed with heavy tails, the $\alpha$-stable distribution is frequently employed. $\alpha$-stable distributions are more flexible in modelling *e.g.* data with skewed centered distributions. The characteristic function of an $\alpha$-stable distribution is $\varphi_{\mathsf{L}_{\alpha\text{-stable}}}(t) = \exp(\mathrm{i}\,t\,\mu - |ct|^\alpha (1 - \mathrm{i}\beta\mathrm{sgn}(t)\Phi))$, where $\mathrm{sgn}(t)$ takes the sign of $t$ and $\Phi = \begin{cases} \tan(\pi\alpha/2) & \alpha \neq 1 \\ -2/\pi \log|t| & \alpha = 1 \end{cases}$. The parameters in $\alpha$-stable distributions are the stability parameter $\alpha$, the skewness parameter $\beta$, the scale parameter $c$, and the location parameter $\mu$. Despite its modelling power, $\alpha$-stable distribution is never employed in PCs, as it is represented analytically by its CF and in most cases does not have a closed-form probability density function.

### 4.1 Theoretic Properties of Characteristic Circuits

With the CC defined above, we can now derive the densities, marginals, and moments from it.

#### 4.1.1 Efficient computation of densities

Through their recursive nature, CCs enable efficient computation of densities in high-dimensional settings even if the density function is not available in closed form. For this, we present an extension of Theorem 3.1 for CCs, formulated using the notion of induced trees $\mathcal{T}$ [Zhao et al., 2016]. A detailed definition of induced trees can be found in Appendix A.2.

**Lemma 4.2** (Inversion). *Let $\mathcal{C} = \langle\mathcal{G}, \psi, \theta_\mathcal{G}\rangle$ be a characteristic circuit on RVs $\boldsymbol{X} = \{X_j\}_{j=1}^d$ with univariate leave nodes. If $\int_\mathbb{R} |\varphi_\mathsf{L}(t)|\mathrm{d}t < \infty$ for every $\mathsf{L} \in V(\mathcal{G})$, then $\boldsymbol{X}$ has a continuous*

*probability density function $f_{\boldsymbol{x}}$ given by $f_{\boldsymbol{X}}(\boldsymbol{x}) =$*

$$\frac{1}{(2\pi)^d} \sum_{i=1}^{\tau} \prod_{(\mathsf{S},\mathsf{N}) \in \mathrm{E}(\mathcal{T}_i)} w_{\mathsf{S},\mathsf{N}} \prod_{\mathsf{L} \in \mathrm{V}(\mathcal{T}_i)} \int_{\mathbb{R}} \exp(-\mathrm{i} t x_{\psi(\mathsf{L})}) \varphi_{\mathsf{L}}(t)\, \mathrm{d}t, \tag{9}$$

*and can be computed efficiently through analytic or numerical integration at the leaves.*

*Proof.* Let $\mathcal{C} = \langle \mathcal{G}, \psi, \theta_{\mathcal{G}} \rangle$ be a characteristic circuit on RVs $\boldsymbol{X} = \{X_j\}_{j=1}^{d}$ with univariate leave nodes and $p_{\mathsf{N}}$ the number of RVs in the scope of $\mathsf{N}$. In order to calculate the density function of $\mathcal{C}$, we need to integrate over the $d$-dimensional real space $\mathbb{R}^d$, *i.e.*,

$$f_{\boldsymbol{X}}(\boldsymbol{x}) = \frac{1}{(2\pi)^d} \underbrace{\int_{\boldsymbol{t} \in \mathbb{R}^d} \exp(-\mathrm{i}\,\boldsymbol{t}^\top \boldsymbol{x})\, \varphi_{\mathcal{C}}(\boldsymbol{t})\, \lambda_d(\mathrm{d}\boldsymbol{t})}_{=\hat{f}_{\mathcal{C}}(\boldsymbol{x})}, \tag{10}$$

where $\varphi_{\mathcal{C}}(\boldsymbol{t})$ denotes the CF defined by the root of the characteristic circuit and $\lambda_d$ is the Lebesque measure on $(\mathbb{R}^d, \mathcal{B}(\mathbb{R}^d))$. We can examine the computation of Eq. (10) recursively for every node.

**Leaf Nodes.** If $\mathsf{N}$ is a leaf node $\mathsf{L}$, we obtain $\hat{f}_{\mathsf{N}}(\cdot)$ by calculating:

$$\hat{f}_{\mathsf{L}}(x) = 2\pi f_{\mathsf{L}}(x) = \int_{\mathbb{R}} \exp(-\mathrm{i} t x) \varphi_X(t) \lambda(\mathrm{d}t), \tag{11}$$

which follows from Theorem 3.1.

**Sum Nodes.** If $\mathsf{N}$ is a sum node $\mathsf{S}$, then:

$$\hat{f}_{\mathsf{S}}(\boldsymbol{x}) = \int_{\boldsymbol{t} \in \mathbb{R}^p} \exp(-\mathrm{i}\,\boldsymbol{t}^\top \boldsymbol{x})\, \varphi_{\mathsf{S}}(\boldsymbol{t})\, \lambda_p(\mathrm{d}\boldsymbol{t}) = \sum_{\mathsf{N} \in \mathrm{ch}(\mathsf{S})} w_{\mathsf{S},\mathsf{N}} \underbrace{\int_{\boldsymbol{t} \in \mathbb{R}^{p_\mathsf{S}}} \exp(-\mathrm{i}\,\boldsymbol{t}^\top \boldsymbol{x})\, \varphi_{\mathsf{N}}(\boldsymbol{t})\, \lambda_{p_\mathsf{S}}(\mathrm{d}\boldsymbol{t})}_{=\hat{f}_{\mathsf{N}}(\boldsymbol{x})}. \tag{12}$$

Therefore, computing the inverse for $\mathsf{S}$ reduces to inversion at its children.

**Product Nodes.** If $\mathsf{N}$ is a product node $\mathsf{P}$, then:

$$\hat{f}_{\mathsf{P}}(\boldsymbol{x}) = \int_{\boldsymbol{t} \in \mathbb{R}^{p_\mathsf{P}}} \exp(-\mathrm{i}\,\boldsymbol{t}^\top \boldsymbol{x})\, \varphi_{\mathsf{P}}(\boldsymbol{t})\, \lambda_{p_\mathsf{P}}(\mathrm{d}\boldsymbol{t}) = \prod_{\mathsf{N} \in \mathrm{ch}(\mathsf{P})} \underbrace{\int_{\boldsymbol{s} \in \mathbb{R}^{p_\mathsf{N}}} \exp(-\mathrm{i}\,\boldsymbol{s}^\top \boldsymbol{x}_{[\psi(\mathsf{N})]}) \varphi_{\mathsf{N}}(\boldsymbol{s})\, \lambda_{p_\mathsf{N}}(\mathrm{d}\boldsymbol{s})}_{=\hat{f}_{\mathsf{N}}(\boldsymbol{x}_{[\psi(\mathsf{N})]})}, \tag{13}$$

where we used that $\lambda_{p_\mathsf{P}} = \otimes_{\mathsf{N} \in \mathrm{ch}(\mathsf{P})} \lambda_{p_\mathsf{N}}$ is a product measure on a product space, applied Fubini's theorem [Fubini, 1907], and used the additivity property of exponential functions. Consequently, computing the inverse for $\mathsf{P}$ reduces to inversion at its children.

Through the recursive application of Eq. (12) and Eq. (13), we obtain that Eq. (10) reduces to integration at the leaves and, therefore, can be solved either analytically or efficiently through one-dimensional numerical integration. $\qquad\square$

### 4.1.2 Efficient computation of marginals

Similar to PCs over distribution functions, CCs allow efficient computation of arbitrary marginals. Given a CC on RVs $\boldsymbol{Z} = \boldsymbol{X} \cup \boldsymbol{Y}$, we can obtain the marginal CC of $\boldsymbol{X}$ as follows. Let $n = |\boldsymbol{X}|$, $m = |\boldsymbol{Y}|$ and let the characteristic function of the circuit be given by

$$\varphi_{\mathcal{C}}(t_1, \ldots, t_n, t_{n+1}, \ldots, t_{n+m}) = \int_{\boldsymbol{z} \in \mathbb{R}^{n+m}} \exp(\mathrm{i}\boldsymbol{t}^\top \boldsymbol{z}) \mu_{\mathsf{S}}(\mathrm{d}\boldsymbol{z}), \tag{14}$$

where $\mu_{\mathsf{S}}$ denotes the distribution of the root. Then the marginal CC of $\boldsymbol{X}$ is given by setting $t_j = 0$, $n < j \leq n + m$. The proof of marginal computation is provided in Appendix B.1.

### 4.1.3 Efficiently computing moments via differentiation

Characteristic circuits also allow efficient computation of moments of distributions. Let $k \in \mathbb{N}^+$ be such that the partial derivative $\frac{\partial^{dk}\varphi_{\mathcal{C}}(\boldsymbol{t})}{\partial t_1^k \cdots \partial t_d^k}$ exists, then the moment $\mathbb{M}_k$ exists and can be computed efficiently through the derivative at the leaves

$$\mathbb{M}_k = \mathrm{i}^{-dk} \frac{\partial^{dk}\varphi_{\mathcal{C}}(\boldsymbol{t})}{\partial t_1^k \cdots \partial t_d^k}\bigg|_{t_1=0,\dots,t_d=0} = \mathrm{i}^{-dk} \sum_{i=1}^{\tau} \prod_{(\mathsf{S},\mathsf{N}) \in \mathrm{E}(\mathcal{T}_i)} w_{\mathsf{S},\mathsf{N}} \prod_{\mathsf{L} \in \mathrm{V}(\mathcal{T}_i)} \frac{\mathrm{d}^k \varphi_{\mathsf{L}}(t_{\psi(\mathsf{L})})}{\mathrm{d}t_{\psi(\mathsf{L})}^k}\bigg|_{t_{\psi(\mathsf{L})}=0}. \quad (15)$$

A detailed proof can be found in Appendix B.2.

## 4.2 Learning Characteristic Circuits from Data

To learn a characteristic circuit from data there are several options. The first option is *parameter learning using a random circuit structure*. The random structure is initialized by recursively creating mixtures with random weights for sum nodes and randomly splitting the scopes for product nodes. A leaf node is created with randomly initialized parameters when there is only one scope in a node. Maximising the likelihood at the root of a CC requires one to apply the inversion theorem to the CC for each training data. When a leaf node does not have a closed-form density function, numerical integration could be used to obtain the density value given data, which makes the maximum likelihood estimation (MLE) at the root not guaranteed to be tractable.

As discussed in prior works, see *e.g.* Yu [2004], minimising a distance function to the ECF is most related to moment-matching approaches, but can result in more accurate fitting results. Therefore, minimising the CFD to the ECF can be beneficial if no tractable form of the likelihood exists but evaluating the characteristic function is tractable. In this case, instead of maximising the likelihood from CC, which is not guaranteed to be tractable, we take the ECF from data as an anchor and minimise the CFD between the CC and ECF:

$$\frac{1}{k}\sum_{j=1}^{k}\left|\frac{1}{n}\sum_{i=1}^{n}\exp(\mathrm{i}\boldsymbol{t}_j^\top \boldsymbol{x}_i) - \varphi_{\mathcal{C}}(\boldsymbol{t}_j)\right|^2. \quad (16)$$

Applying Sedrakyan's inequality [Sedrakyan, 1997] to Eq. (16), parameter learning can be operated batch-wise:

$$\frac{1}{k}\sum_{j=1}^{k}\left|\frac{1}{b}\sum_{l=1}^{b}\frac{1}{n_b}\sum_{i=1}^{n_b}\exp(\mathrm{i}\boldsymbol{t}_j^\top \boldsymbol{x}_i) - \varphi_{\mathcal{C}}(\boldsymbol{t}_j)\right|^2 \leq \frac{1}{k}\sum_{j=1}^{k}\frac{1}{b}\sum_{l=1}^{b}\left|\frac{1}{n_b}\sum_{i=1}^{n_b}\exp(\mathrm{i}\boldsymbol{t}_j^\top \boldsymbol{x}_i) - \varphi_{\mathcal{C}}(\boldsymbol{t}_j)\right|^2,$$

where $b$ is the number of batches and $n_b$ the batch size. This way, parameter learning of CC is similar to training a neural network. Furthermore, if two CCs are compatible, as similarly defined for PCs [Vergari et al., 2021], the CFD between the two CCs can be calculated analytically, see Appendix D for more details.

---

**Algorithm 1** CC Structure Learning

**Input:** training data $\mathcal{D}$, RVs $\boldsymbol{X}$, $min\_k$, $k_{\mathsf{S}}$, $k_{\mathsf{P}}$
**Output:** $\mathcal{C}$
**Function** buildCF($\mathcal{D}$, $X$)
    return $\mathsf{L} \leftarrow$ *univariate CF* $\varphi_X(t)$

**Function** buildSumNode($\mathcal{D}$, $\boldsymbol{X}$)
    **if** $|\boldsymbol{X}| = 1$ **then**
        $\mathsf{S} \leftarrow$ buildCF($\mathcal{D}$, $X$)
    **else if** $|\mathcal{D}| \leq min\_k$ **then**
        Partition $\mathcal{D}$ into $|\boldsymbol{X}|$ independent subsets $\mathcal{D}_j$ ;
        $\mathsf{S} \leftarrow \prod_{j=1}^{|\boldsymbol{X}|}$ buildCF($\mathcal{D}_j$, $X_j$) ;
    **else**
        Partition $\mathcal{D}$ into $k_{\mathsf{S}}$ clusters $\mathcal{D}_i$ ;
        $\mathsf{S} \leftarrow \sum_{i=1}^{k_{\mathsf{S}}} \frac{|\mathcal{D}_i|}{|\mathcal{D}|}$ buildProdNode($\mathcal{D}_i$, $\boldsymbol{X}$)
    return $\mathsf{S}$
**Function** buildProdNode($\mathcal{D}$, $X$)
    Partition $\mathcal{D}$ into $k_{\mathsf{P}}$ independent subsets $\mathcal{D}_j$ ;
    return $\mathsf{P} \leftarrow \prod_{j=1}^{k_{\mathsf{P}}}$ buildSumNode($\mathcal{D}_j$, $\boldsymbol{X}_j$)
$\mathcal{C} \leftarrow$ buildSumNode($\mathcal{D}$, $\boldsymbol{X}$)

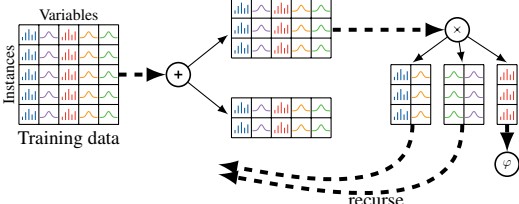

Figure 2: Illustration of the recursive structure learning algorithm. Sum nodes are the result of clustering, having weighted children that are product nodes. Product nodes are the result of independence test, enforcing independence assumptions of their children. Leaf nodes are univariate characteristic functions modelling local data.

Table 1: Average test log-likelihoods from CC after parameter learning by minimising the CFD on synthetic data sets. The CC structure is either generated using Random Structure or learned using the Structure Learning algorithm.

| Data Set | Random Structure | Random Structure & Parameter Learning | Structure Learning | Structure Learning & Parameter Learning | Structure Learning (random $\boldsymbol{w}$) & Parameter Learning |
|---|---|---|---|---|---|
| MM | -4.93 | -3.50 | -2.87 | -2.86 | -3.34 |
| BN | -6.30 | -4.12 | -3.27 | -3.27 | -3.93 |

However, relying on randomly initialized structures (*e.g.*, due to the fixed split of scopes) may also limit the performance of parameter learning of CC. To overcome this, we derive now a *structure learning* algorithm to learn the structure of the CC. Inspired by Gens and Domingos [2013], this structure learning recursively splits the data slice and creates sum and product nodes of the CC as summarized in Algorithm 1 and depicted in Fig. 2. To create a sum node S, clustering algorithms, *e.g.*, K-means clustering, is employed to split data instances in the slice into $k_{\mathsf{S}}$ subsets. The weights of the sum node are then determined by the portion of data instances in each subset. To create a product node P, some independence tests—*e.g.*, G-test of independence or random dependency coefficient (RDC) based splitting [Molina et al., 2018]—are used to decide on splitting the random variables in the data slice into $k_{\mathsf{P}}$ sub-groups. The sum and product nodes are created recursively until any of the following conditions fulfils: (1) There is only one scope in the data slice, and then a leaf node with the corresponding scope is created. (2) The number of data instances in the data slice is smaller than a pre-defined threshold $min\_k$. In the latter case, a naive factorization is applied to the scopes in the data slice to create a product node, and then create leaves for each scope as children for this product node. When creating a leaf node, the leaf parameters are estimated by MLE if the closed-form density function is available. In the case of ECF leaves, the leaf nodes are created from local data following the definition of ECF in Eq. (4). When there is no closed-form density at a leaf, the parameters of an $\alpha$-stable distribution are estimated using the algorithm in McCulloch [1986].

## 5 Experimental Evaluation

Our intention here is to evaluate the performance of characteristic circuit on synthetic data sets and UCI data sets, consisting of heterogeneous data. The likelihoods were computed based on the inversion theorem. For discrete and Gaussian leaves, the likelihoods were computed analytically. For $\alpha$-stable leaves, the likelihoods were computed via numerical integration using the Gauss-Hermit quadrature of degree 50.

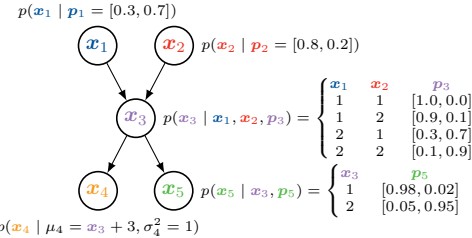

Figure 3: The Bayesian network used for BN.

**Can characteristic circuits approximate known distributions well?** We begin by describing and evaluating the performance of CC on two synthetic data sets. The first data set consisted of data generated from a mixture of multivariate distributions (denoted as MM): $p(\boldsymbol{x}) = \sum_{i=1}^{K} w_i p(\boldsymbol{x}_1 \mid \mu_i, \sigma_i^2) p(\boldsymbol{x}_2 \mid \boldsymbol{p}_i)$, where $p(\boldsymbol{x} \mid \mu, \sigma^2)$ is the univariate normal distribution with mean $\mu$ and variance $\sigma^2$, and $p(\boldsymbol{x} \mid \boldsymbol{p})$ is the univariate categorical distribution with $\boldsymbol{p}$ the vector of probability of seeing each element. In our experiments we set $K = 2$ and $w_1 = 0.3$, $w_2 = 0.7$. For each univariate distribution we set $\mu_1 = 0$, $\sigma_1^2 = 1$, $\mu_2 = 5$, $\sigma_2^2 = 1$, $\boldsymbol{p}_1 = [0.6, 0.4, 0.0]$ and $\boldsymbol{p}_2 = [0.1, 0.2, 0.7]$. The second data set consisted of data generated from a Bayesian network with 5 nodes (denoted as BN), to test the modelling power of characteristic circuits with more RVs and more complex correlations among each RV. The details of the BN are depicted in Fig. 3. Here, $\boldsymbol{X}_1, \boldsymbol{X}_2, \boldsymbol{X}_3$ and $\boldsymbol{X}_5$ are binary random variables parameterized by $\boldsymbol{p}_i$, and $\boldsymbol{X}_4$ is a continuous random variable conditioned on $\boldsymbol{X}_3$. For both data sets MM and BN, 800 instances were generated for training and 800 for testing.

We first employed parameter learning and evaluated the log-likelihoods from the random structure and after parameter learning. A detailed setting of parameter learning is illustrated in Appendix C.1. The increase of log-likelihoods after parameter learning (columns 2 and 3 in Table 1) implies that

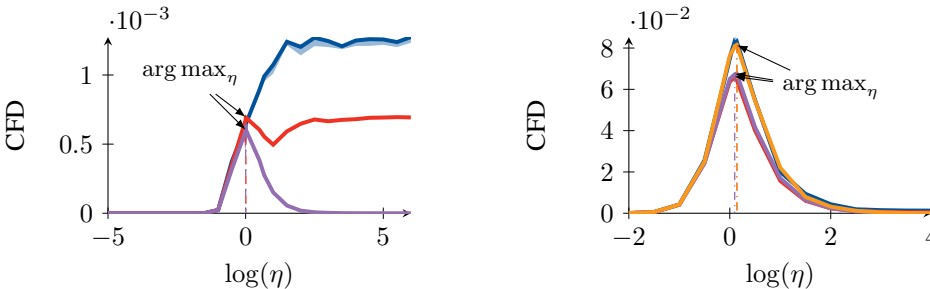

Figure 4: Characteristic circuits approximate the true distributions better than the ECF by providing a smaller CFD. We visualize the CFD for CC with parametric leaves (CC-P ▬), ECF as leaves (CC-E ▬), normal distribution as leaves (CC-N ▬) and a single empirical characteristic function (ECF ▬) learned from synthetic heterogeneous data (Left: MM, Right: BN). Best viewed in color.

minimising the CFD pushes CC to better approximate the true distribution of data. We then learnt CCs using structure learning with $min\_k = 100$, and $k_S = k_P = 2$. Various leaf types were evaluated: CC with ECF as leaves (CC-E), CC with normal distribution for continuous RVs and categorical distributions for discrete RVs, *i.e.*, parametric leaves (CC-P), and CC with normal distribution for all leaf nodes (CC-N). The trained CCs were evaluated with the CFD between the CC and the ground truth CF. For data set BN, the ground truth CF was derived via the algorithm for generating arithmetic circuits that compute network polynomials [Darwiche, 2003]. Following Chwialkowski et al. [2015] and Ansari et al. [2020], we illustrate both the CFD with varying scale $\eta$ in $\omega(\boldsymbol{t}; \eta)$ and also optimising $\eta$ for the largest CFD, shown in Fig. 4. We report average CFD values and standard deviations obtained from five runs. It can be seen from Fig. 4 that both CC-E and CC-P have almost equally lower CFD values and also lower maximum CFD values compared to the ECF, which indicates the characteristic circuit structure better encodes the data distribution than the ECF. The smaller standard deviation values from characteristic circuits compared with ECF also imply that characteristic circuits offer a more stable estimate of the characteristic function from data. For data set MM, the maximum CFD of CC-N is 0.0270 when $\log(\sigma) = 0.6735$, which is far more than 0.0006 of CC-P, and thus not visualized in Fig. 4 (Left). This also happens to data set BN, as can be seen in Fig. 4 (Right) that CC-N gives higher CFD than CC-P and CC-E, which implies that assuming a discrete RV as Normal distributed is not a suitable choice for CC. In addition, parameter learning on CCs from structure learning and structure learning with randomized parameters (last 2 columns in Table 1) provides higher log-likelihoods than random structures, which implies a well-initialized structure improves parameter learning. To conclude, CC estimates the data distribution better than ECF, which is justified by the smaller CFD from CC-E compared with ECF.

**Can characteristic circuits be better density estimators on heterogeneous data?** Real-world tabular data usually contain both discrete and real-valued elements, and thus are in most cases heterogeneous. Therefore, we also conducted density estimation experiments on real-world hetero-geneous data sets and compared to state-of-the-art probabilistic circuit methods, including Mixed SPNs (MSPN) [Molina et al., 2018], Automatic Bayesian Density Analysis (ABDA) [Vergari et al., 2019] and Bayesian SPNs (BSPN) [Trapp et al., 2019]. We employed the heterogeneous data from the UCI data sets, see Molina et al. [2018] and Vergari et al. [2019] for more details on the data sets. Similar to the setup in Trapp et al. [2019], a random variable was treated as categorical if less than 20 unique states of that RV were in the training set. All the rest RVs were modelled with either normal distributions (CC-P) or $\alpha$-stable distributions (CC-A). Again, we first employed parameter learning with a (fixed) random structure using $\alpha$-stable distribution for continuous RVs and report the log-likelihoods (Parameter Learning in Table 2). Note that $\alpha$-stable distributions can not be represented with closed-form densities, thus maximising the likelihood from it can not be solved exactly and efficiently. As a comparison, structure learning was also employed with $min\_k = 100$, $k_S = k_P = 2$ for G-test based splitting (CC-P & CC-A), and with $min\_k = 100$, $k_S = 2$ for RDC based splitting (CC-A$^{\text{RDC}}$). A detailed description of the experimental settings can be found in Appendix C.2. The test log-likelihoods are presented in Table 2. As one can see, parameter learning performs worse than CC-A but still outperforms some of the baselines. CC-P does not win on all the data sets but is competitive with MSPN and ABDA on most of the data sets. CC-A outperforms the baselines on 8 out of 12 data sets, and CC-A$^{\text{RDC}}$ outperforms all the other methods

Table 2: Average test log-likelihoods from CC and SOTA algorithms on heterogeneous data.

| Data Set | Parameter Learning | MSPN | ABDA | BSPN | Structure Learning | | |
|---|---|---|---|---|---|---|---|
| | | | | | CC-P | CC-A | CC-A$^{RDC}$ |
| Abalone | 3.06 | 9.73 | 2.22 | 3.92 | 4.27 | _15.10_ | **17.75** |
| Adult | -14.47 | -44.07 | -5.91 | _-4.62_ | -31.37 | -7.76 | **-1.43** |
| Australian | -5.59 | -36.14 | -16.44 | -21.51 | -30.29 | _-3.26_ | **-2.94** |
| Autism | -27.80 | -39.20 | -27.93 | **-0.47** | -34.71 | -17.52 | _-15.5_ |
| Breast | -20.39 | -28.01 | -25.48 | -25.02 | -54.75 | _-13.41_ | **-12.36** |
| Chess | -13.33 | -13.01 | _-12.30_ | **-11.54** | -13.04 | -13.04 | -12.40 |
| Crx | -6.82 | -36.26 | -12.82 | -19.38 | -32.63 | _-4.72_ | **-3.19** |
| Dermatology | -45.54 | -27.71 | -24.98 | _-23.95_ | -30.34 | -24.92 | **-23.58** |
| Diabetes | -1.49 | -31.22 | -17.48 | -21.21 | -23.01 | **0.63** | _0.27_ |
| German | -19.54 | -26.05 | -25.83 | -26.76 | -27.29 | _-15.24_ | **-15.02** |
| Student | -33.13 | -30.18 | -28.73 | -29.51 | -31.59 | _-27.92_ | **-26.99** |
| Wine | 0.32 | -0.13 | -10.12 | -8.62 | -6.92 | _13.34_ | **13.36** |
| # best | 0 | 0 | 0 | 2 | 0 | 1 | 9 |

on 9 out of 12 data sets. This implies that characteristic circuit, especially with structure learning, is a competitive density estimator compared with SOTA PCs. Actually, $\alpha$-stable leaf distributions are a more suitable choice for characteristic circuits on heterogeneous tabular data.

## 6  Conclusion

We introduced characteristic circuits (CCs), a novel circuit-based characteristic function estimator that leverages an arithmetic circuit with univariate characteristic function leaves for modelling the joint of heterogeneous data distributions. Compared to existing PCs, characteristic circuits model the characteristic function of data distribution in the continuous spectral domain, providing a unified view for discrete and continuous random variables, and can further model distributions that do not have closed-form probability density functions. We showed that both joint and marginal probability densities can be computed exactly and efficiently using characteristic circuits. Finally, we empirically showed that characteristic circuits approximate the data distribution better than ECF, measured by the squared characteristic function distance, and that characteristic circuits can also be competitive density estimators as they win on 9 out of 12 heterogeneous data sets compared to SOTA models.

There are several avenues for future work. For instance, sampling from characteristic functions and, in turn, characteristic circuits is not straightforward. One should explore existing literature discussing sampling from CFs [Devroye, 1986, Ridout, 2009, Walker, 2017], and adapt them to sampling from CCs. The circuit structure of characteristic circuits generated by structure learning has a high impact on the performance of the characteristic circuit, and therefore an inappropriate structure can limit the modelling power of characteristic circuits. Therefore, one should explore parameter learning of characteristic circuits on more advanced circuit structures [Peharz et al., 2020] and, in particular, using normalizing flows, resulting in what could be called characteristic flows.

**Broader Impact.** Our contributions are broadly aimed at improving probabilistic modelling. CCs could be used to develop more scalable and more accurate probabilistic models, in particular over mixed domains as common in economics, social science, or medicine. Scaling to even bigger mixed models can open up even more potential applications, but also may require careful design to handle overconfidence and failures of CC.

## Acknowledgments and Disclosure of Funding

This work was supported by the Federal Ministry of Education and Research (BMBF) Competence Center for AI and Labour ("KompAKI", FKZ 02L19C150). It benefited from the Hessian Ministry of Higher Education, Research, Science and the Arts (HMWK; projects "The Third Wave of AI" and "The Adaptive Mind"), and the Hessian research priority program LOEWE within the project "WhiteBox". MT acknowledges funding from the Academy of Finland (grant number 347279).

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
