# OpenReview forum: "Characteristic Circuits"
_NeurIPS.cc/2023/Conference — NeurIPS 2023 oral_

### Official Review · Reviewer_6Se4 · 2023-06-08

**Soundness:** 4 excellent
**Presentation:** 4 excellent
**Contribution:** 4 excellent
**Rating:** 10
**Confidence:** 1

**Summary:**

I am not qualified to review this paper.

**Strengths:**

I am not qualified to review this paper.

**Weaknesses:**

I am not qualified to review this paper.

**Questions:**

I am not qualified to review this paper.

**Limitations:**

I am not qualified to review this paper.

---

### Official Review · Reviewer_twZC · 2023-06-20

**Soundness:** 3 good
**Presentation:** 4 excellent
**Contribution:** 2 fair
**Rating:** 5
**Confidence:** 4

**Summary:**

This work proposes a new tractable probabilistic model called characteristic circuits or CC. The CC is defined in a similar way to probabilistic circuits (PCs) but with leave nodes defined as characteristic functions instead of the distributions as in PCs. The authors further show the computation of marginals and the learning algorithms for CC. Empirical evaluations on both synthetic datasets and UCI datasets are presented.

**Strengths:**

- This paper is generally well-written, with sufficient backgrounds provided to help readers understand the proposed new model.
- The authors show that CC shares the same efficient marginal computations as in PCs.
- CC provides a unified view for mixed continuous and discrete distributions. Also, the \alpha-stable distributions are less explored in the previous literature of tractable probabilistic models to my knowledge and the use of these seems help deliver good performance in the UCI experiments.

**Weaknesses:**

- From what is presented, it seems that the authors simply rewrite the distribution nodes and computations of PCs into their characteristic function duals. It is unclear what are the key differences between CC and PCs and when would one prefer one over the other. I don't think the unified view of the discrete and continuous distributions serves as a strong motivation since mixed SPN can also handle the mixed distributions.
- To further illustrate the previous point, one would expect to see investigations on expressiveness, such as are there any distributions that can be tractably represented by CCs but not PCs, or investigations on tractable operations, such as what probabilistic queries would be intractable for PC but tractable for CC, while none of these are discussed in this work.
- A proof for the validity of the marginal computations seems to be missing.

**Questions:**

- Can the authors elaborate on the motivation for CCs, that is, when one would prefer CC over PC?
- Can the authors provide some discussions on the expressiveness and tractable queries of CC?
- For the empirical results on UCI datasets, I wonder if the improvements are from CC itself or from the alpha-stable distributions. What would happen if the continuous leaves in MSPN are defined as alpha-stable distributions? Such an ablation study might make the results more convincing.

**Limitations:**

Yes.

---

> ### Author Rebuttal · Authors · 2023-08-09
>
> We would like to thank the reviewer for the feedback and questions.
>
> > Key difference between CCs and PCs:
>  - PCs do not naturally lend themselves to a unified view over heterogeneous data domains, while CCs more naturally provide a framework to model high-dimensional mixed data distributions. To highlight this, recall that PCs are formulated over density/mass functions which is the Radon–Nikodym derivative w.r.t. some base measure. In heterogeneous domains, the relationship to the base measure becomes involved as dimensions might have a density w.r.t. different base measures. This fact is typically hidden in PCs and results in challenges when learning such models over heterogeneous data domains as, for example, gradient-based parameter learning will now depend on different base measures depending on the dimensions considered. This can result in undesirable behaviour and is elevated, for example, in MSPNs by discretizing the continuous domain, hence, ensuring the same base measure for all dimensions. However, discretization introduces challenges going beyond this rebuttal. CCs, on the other hand, evaluate this problem in a more principled way by instead modelling the Fourier transform of the probability measure directly. This makes the learning and modelling process independent of the base measure and provides a truly unified view compared to PCs. We hope that this expose better explains the conceptual difference between CCs and PCs.
>
> > Expressiveness and tractable queries:
>  - We agree that investigating the expressive efficiency and tractability of CCs is an interesting avenue. Although not discussed in detail in the paper, the moments of any CC can be computed tractably through differentiation given that the leaf nodes allow tractable differentiation. Even though the current work focuses on a unified view by moving to the spectral domain, we believe this to be a particularly useful property and added further details to the updated manuscript. We leave a theoretical analysis of the expressive efficiency of CCs for future work.
>
> > Proof of marginal computation:
>  - A proof sketch of marginalization is given in lines 221 to 225 and we will add a more detailed proof in the appendix.
>  - Proof sketch:
> Following lines 216 to 225, we have assumed that the circuit decomposes all dimensions into univariate leaves, RVs $Z = X \cup Y$, $t = t_X \cup t_Y$, and we aim to compute $\varphi_X(t_X)$.
> Then for any leaf node, we have
> $\varphi_L(t_j) =  1$ if $ t_j = 0$, and  $\varphi_L(t_j)$ otherwise, by definition of CFs.
> Let $P$ be a product node that splits at least one $Y_j$ from its scope into a single child and let this child be denoted as $N_j$, then
> $\varphi_P(t \cup 0) = \varphi_{N_j}(0) \prod_{N \in ch(P)\setminus{N_j}} \varphi_{N}(t_{\psi(N)}) = \varphi_{P\setminus{N_j}}(t)$,
> where $\varphi_{N_j}(0)=1$.
> By the assumption that sum nodes are convex combinations (weights sum up to one) and recursive application of the above, one can show that any marginal distribution can be obtained tractably in CCs.
>
> > Choice of CC over PC:
>
>  - There are various tasks in which CCs are preferable over PCs. In particular, as outlined before, CCs provide a more principled and natural representation in the case of heterogeneous data domains. Moreover, CCs enable the modeling and learning of distributions that do not have an analytical density function. Lastly, CCs provide an efficient representation of moments even in case densities are not available in closed form as CCs circumvent the challenge of integration in this case and instead only require differentiation of the model.
>
> > $\alpha$-stable distribution leaves and MSPNs:
>
>  - We agree that studying the combinations of MSPNs and CCs is an interesting direction. Henceforth, we will provide additional results using MSPNs as a construction algorithm for the CC structure. However, we want to stress that MSPNs and CCs are conceptually very different as MSPNs aim to model heterogeneous domains non-parametrically through discretization, while CCs directly model the characteristic function of the mixed distribution. Therefore, CCs provide a more flexible framework and allow for meaningful parameter learning that is more suited to mixed data domains. Moreover, we want to note that we empirically observed improvements by fitting CCs (work in the spectral domain) also in the case $\alpha$-stable distribution has not been employed, see results CC-P in Tab. 2, indicating that CCs are a promising modeling family even if tractable densities exist.

---

> > ### Comment · Reviewer_twZC · 2023-08-16
> >
> > I would like to thank the authors for the clarification. I'm happy to raise my score.

---

### Official Review · Reviewer_HuCd · 2023-06-28

**Soundness:** 3 good
**Presentation:** 3 good
**Contribution:** 3 good
**Rating:** 6
**Confidence:** 4

**Summary:**

The paper introduces characteristic circuits (CCs), a new family of tractable probabilistic models (TPMs) that leverages univariate characteristic functions as leaves of probabilistic circuits (PCs) for modelling a tractable joint of heterogeneous data distributions (i.e. with both continuous and discrete variables).
CCs model the characteristic function of the data distribution in the continuous spectral domain (cf. Equation 1), thus providing a unified framework for discrete and continuous random variables.
As a consequence, one of the main advantage of CCs is that they can model distributions that do not have closed-form probability density functions, such as $\alpha$-stable distributions.
Importantly, authors also show that CCs allow exact and efficient computation of joint and marginal probabilistic queries.
CCs are evaluated on two synthetic datasets and 12 heterogeneous real-world tabular datasets.


**Strengths:**

- The research is definitely original as it proposes a new class of TPMs with many (novel) benefits
- The model naturally lends itself to modelling heterogeneous data
- The model allows to use distributions that do not have closed-form expressions, such as $\alpha$-stable distributions, something that is not possible in current PCs
- Despite having input units with no closed-form expressions, CCs can still deliver exact marginalisation

Overall, a solid contribution.



**Weaknesses:**

- From experiments, it looks like inappropriate structure can limit the modelling power of CCs. This is can prevent using CCs when a good structure is not available.
- It's unclear how reliable/precise numerical integration can be (lines 128-129-130)
- Sampling it's an important inference routine of PCs yet it is not discussed at all, and it's unclear if CCs can provide it

**Questions:**

- Is sampling possible from CCs? If yes, can we know how CC samples compare with the ones of standard PCs? If not, what are the challenges?
- Can you elaborate a bit more on what precisely you mean with "unified view for discrete and continuous random variables" (line 57)? While, to some extent, I understand what author mean, one may think "but even standard PCs can handle heterogeneous data". I think being more precise here can improve this major selling point of the paper.
- Can you elaborate a bit more on lines 230-231-232? Why MLE is not tractable? How does Eq. 14 relate to MLE? (Also lines 248-249-250 are unclear to me)


**Limitations:**

Limitations are not explicitly addressed.
It looks like sampling can be one of these.

Minors:
- In Lemma 4.2, I think there's no explanation of what $\tau$ and $E(\mathcal{T}_i)$ represent. I know it's notation related to induced trees, but it is not introduced in the text.
- In line 175, there's no $x$ occuring in the definition of $\phi_{L_\text{Normal}}(t)$, why?

---

> ### Author Rebuttal · Authors · 2023-08-09
>
> We would like to thank the reviewer for pointing out both the strength and possible weakness of our work.
>
> > Inappropriate structure can limit the modelling power:
>
>  - Similar to related modelling families (e.g., PCs, PGCs), the structure can have a high impact on the performance of the CC. To mitigate this issue, we proposed the first structure learning algorithm adapted from the well-known algorithm by [Gens and Pedro, 2013] for PCs. Note that structure learning of circuits (PCs, CCs alike) is a challenging and open task and further investigation is needed. However,  an approach similar to the one taken in Einsum networks [Peharz et al. 2020] combined with minimization of the CFD could be a promising future direction.
>
> > Reliability of numerical integration:
>
>  - We ran additional experiments with an increasing number of sample points to verify the reliability of the numerical integration through quadrature. The results indicate that a low number of sample points is sufficient as numerical integration is only required on the real line (1D). We thank the reviewer for pointing this out and will include the results into the Appendix and add further discussion on the reliability of the numerical integration.
>
> > Sampling:
>
>  - Sampling from a characteristic function is generally not straightforward. There has been literature discussing sampling from CFs [Devroye, 1986, Ridout, 2009, Walker, 2017] and we believe these sampling algorithms can be adapted to sampling from CC in future works. Thanks for pointing this out, we will add this to the discussion of interesting future work.
>
> > Unified view:
>
>  - PCs do not naturally provide a unified view and treat discrete and continuous RVs differently. For discrete RVs probabilities or mass values are computed w.r.t. the  counting measure, while for continuous RVs the reference measure is the Lebesgue measure. Moreover, RVs distributed according to a singular (continuous) distribution can typically not be represented at all. This dependence on the base measure is hidden in PCs and can result in challenges when it comes to learning these models in heterogeneous domains. For example, a model might focus only on maximizing the likelihood w.r.t. the Lebesgue measure during fitting. Consequently, prior works have suggested discretising the domain of continuous RV (see MSPNs), which introduces new challenges. Moving away from the dependence on the base measure by representing the distribution through its characteristic function, which is independent of the base measure, elevates this issue. Hence, CCs provide a truly unified view compared to PCs. We will clarify the “unified view” in the revised paper to better reflect our contribution.
>
> > MLE at the root and at a leaf node:
>
>  - In parameter learning, maximizing the likelihood at the root of a CC needs to apply the inversion theorem to CC for each training data. When leaf nodes do not have a closed-form density function, numerical integration has to be employed to obtain the density value given data. This makes the MLE at the root not guaranteed to be tractable.
>  - We thank the reviewer for raising this interesting question about the relationship between minimizing the distance and MLE, which is similar to the question from reviewer 1MK1. Connections between maximum likelihood estimation (MLE) and minimizing a distance (e.g., the CFD) to the empirical characteristic function (ECF) is indeed an interesting question. Minimizing the CFD to the ECF can be beneficial if no tractable form of the likelihood exists but the characteristic function can be tractably evaluated. As discussed in prior works (e.g., In [Yu, 2004]), minimizing a distance function to the ECF is most related to moment-matching approaches, but can result in more accurate fitting results. We will add further detail and a discussion on the topic to the revised version.
>  - Creating leaf nodes in structure learning. Leaf nodes are created by fitting the estimated distribution to local data during structure learning. When closed-form density/mass function is available at a leaf, the leaf parameters can be estimated via MLE. In the case of ECF leaves, the leaf nodes are created from local data following the definition of ECF (in line 134). When there is no closed-form density, e.g. $\alpha$-stable distributions, the algorithm in [McCulloch, 1986] is employed to estimate the parameters at the $\alpha$-stable leaves. We apologise that we did not specify this detail in the manuscript and will add the above to lines 248-250 for better clarification.
>
> > Minors:
>  - Induced trees notation: Thanks for pointing this out, we will add one section in the Appendix to briefly introduce the notation of induced trees.
>  - Indeed there is no x in the definition of the characteristic function at the leaf, because a characteristic function is a function of t, as illustrated in Eq (1).
> ***
> [Gens and Pedro, 2013] Robert Gens and Domingos Pedro. "*Learning the structure of sum-product networks.*" In ICML, 2013.
> [Peharz et al. 2020] Robert Peharz et al. "*Einsum networks: Fast and scalable learning of tractable probabilistic circuits.*" In ICML, 2020.
> [Devroye, 1986] Luc Devroye, "*An automatic method for generating random variates with a given characteristic function.*" SIAM journal on applied mathematics, 1986.
> [Ridout, 2009] Martin S Ridout. "*Generating random numbers from a distribution specified by its Laplace transform.*" Statistics and Computing, 2009.
> [Walker, 2017] Stephen G Walker. "*A Laplace transform inversion method for probability distribution functions.*" Statistics and Computing, 2017.
> [Yu, 2004] Jun Yu. "*Empirical characteristic function estimation and its applications.*" Econometric reviews, 2004.
> [McCulloch, 1986] J. Huston McCulloch. "*Simple consistent estimators of stable distribution parameters.*" Communications in statistics-simulation and computation, 1986.

---

> > ### Comment · Reviewer_HuCd · 2023-08-10
> >
> > Many thanks for your clarifications. I further confirm the positive impression I had, and I'll keep supporting the paper. However, I'd stick to my score: I would have raised my score if sampling had been possible in CCs (at the very least, it's unclear if it is going to be).

---

### Official Review · Reviewer_pe9b · 2023-07-04

**Soundness:** 4 excellent
**Presentation:** 4 excellent
**Contribution:** 3 good
**Rating:** 7
**Confidence:** 3

**Summary:**

This manuscript proposes a framework for directly representing characteristic function of random variables by a probabilistic circuit-like structure. Unlike the ordinal probabilistic networks, the proposed framework, characteristic circuits, can treat distributions that do not have closed-form expressions for the density, or even marginals of discrete and continuous random variables. This is because the characteristic function provides a unified view for both discrete and continuous random variables. As a result, the characteristic circuit can treat broader class of distributions than the ordinal probabilistic circuits. Despite this, it is proved that the characteristic circuit keeps tractability of computing densities and marginals, which is an important query for probabilistic inference. Also, the parameter and circuit-structure learning algorithms for characteristic circuits are given. The experiments showed that the proposed characteristic circuits perform well for density estimation task on heterogeneous data sets, which includes both discrete and continuous variables.

**Strengths:**

I think that representing a characteristic function by a circuit is a simple yet strong idea for representing broader class of distributions. Also, proposing an algorithm for computing marginals on characteristic circuits is really good, because it inherits the strengths of the original probabilistic circuits that some rich inference queries are tractable in time proportional to the size of the circuit; although it only proves the marginal query, I think this query is one of the most important one for probabilistic inference. The experimental results truly support the usefulness of characteristic circuits when applied to density estimation task where the evaluation metric is the test log-likelihood. Since the original probabilistic circuits also show their strengths in this task, I think the selection of tasks is appropriate.

**Weaknesses:**

It is not the first attempt to represent a generating function regarding random variables directly; the first one (as far as I know) is:
Probabilistic Generating Functions https://arxiv.org/abs/2102.09768 (published in ICML 2021).
This represents the probability generating function directly, thus I think this framework is for discrete variables only. However, to clearly show the standing position of this paper, the comparison with this work should be clearly done in the main article.

**Questions:**

Within the queries that are tractable for the ordinal probabilistic circuits, are there any other tractable queries than marginal for characteristic circuits? Or, are there any queries that is proven to be hard (e.g., NP-complete) for characteristic circuits?

**Limitations:**

I think the authors adequately addressed the limitations.

---

> ### Author Rebuttal · Authors · 2023-08-09
>
> We thank the reviewer for the feedback and the suggested related work.
>
> > Comparison to PGCs
>
>  - Indeed, PGCs are related to CCs as both can be considered to represent the probability distribution using its generating function rather than its density function. We added a discussion and further details on how the two approaches relate to the updated manuscript. The key difference between PGCs and CCs is that, while PGCs can only represent discrete probability distributions that admit a probability generating function representation (finite countable), CCs can represent any probability distribution as every probability measure has an associated characteristic function. Interestingly, compared to directly modeling the density/mass function of a distribution, we can perform model fitting even in cases where a density w.r.t. the Lebesgue or counting measure does not exist or is not tractable to evaluate by minimizing the CFD to the empirical characteristic function.
>
> > Are there any other tractable queries for characteristic circuits?
>
>  - Although not discussed in detail in the paper, the moments of any CCs can be computed tractably through differentiation. Even though the current work focuses on a unified view by moving to the spectral domain, we believe this to be a particularly useful property and added further details to the updated manuscript.
>  - Sampling from a characteristic function is generally not straightforward. There has been literature discussing sampling from CFs [Devroye, 1986, Ridout, 2009, Walker, 2017] and we believe these sampling algorithms can be adapted to sampling from CCs in future works. We thank the reviewer for pointing this out and will add a discussion to the manuscript.
> ***
> [Devroye, 1986] Luc Devroye, "*An automatic method for generating random variates with a given characteristic function.*" SIAM journal on applied mathematics, 1986.
> [Ridout, 2009] Martin S Ridout,. "*Generating random numbers from a distribution specified by its Laplace transform.*" Statistics and Computing, 2009.
> [Walker, 2017] Stephen G Walker. "*A Laplace transform inversion method for probability distribution functions.*" Statistics and Computing, 2017.

---

> > ### Comment · Reviewer_pe9b · 2023-08-14
> >
> > Thank you very much for detailed reply. I think the comparison of probabilistic generating circuits and characteristic circuits is adequately addressed in the rebuttal comment.
> > Regarding the tractability of queries, I agree with Reviewer HuCd in that the sampling is one of the most important query that PCs can do. However, I think that even if sampling is generally difficult for CCs, the other parts of the paper (concepts, comparison with PCs, and the empirical results) can constitute significant technical contributions. Thus, I am in favor of accepting this manuscript.

---

### Official Review · Reviewer_1MK1 · 2023-07-07

**Soundness:** 4 excellent
**Presentation:** 3 good
**Contribution:** 4 excellent
**Rating:** 7
**Confidence:** 3

**Summary:**

This paper studies the use of characteristic functions as probabilistic models for heterogeneous data and proposes characteristic circuits (CC) for their representations. The authors propose efficient algorithms for computing (marginal) densities with CCs and show that parameters of CCs can be learned by minimizing the CFD between CC and ECF. The authors also show that CCs achieve strong performance not only on synthetic data but also on some commonly used density estimation benchmarks.

**Strengths:**

To the best of my knowledge, the use of characteristic functions as a new language for probabilistic modeling, especially as a unified framework for heterogeneous data, is very novel. The empirical results is also very strong. I believe this work opens a brand new avenue for density estimation.

**Weaknesses:**

As a non-expert, I spent a lot of time on the background section; despite the uniqueness part in the inversion theorem, it is not completely intuitive how a probability measure is encoded as its characteristic function. It would be helpful if the authors could provide at least one simple example.

Authors use too many acronyms throughout the paper, especially in Section 5; it is not easy for me as a reader to distinguish between CC-N, RS, SL, CFD and etc.

For non-expert readers like me, more details on the empirical evaluations (in the main paper) could be helpful: e.g. how are the likelihoods measured for the heterogeneous datasets? are they computed via numerical integrations? etc.

**Questions:**

In Section 4.2, the authors proposes to do parameter learning by minimizing the CFD between CC and ECF because the likelihood of CC is not guaranteed to be tractable. Yet I wonder if it is possible to discuss the relationship between CFD and likelihood.

**Limitations:**

The structure learning algorithm seems more of an adaptation of the existing structure learning algorithms for SPNs. This is not a major issue as structure learning of circuits has been a very challenging problem and probably not a main focus of this work.

---

> ### Author Rebuttal · Authors · 2023-08-09
>
> We would like to thank the reviewer for the feedback and questions.
>
> > How is the probability measure encoded as a CF?
>
>  - The characteristic function of a probability measure is its Fourier transform and, hence, can be obtained through the application of the Fourier transform. However, in our work, we do not start from the probability measure but directly model the characteristic function instead. This allows us to implicitly learn any probability measure by instead learning its spectral form. We refer to [Sasv&#225;ri, 2013] for a more detailed discussion.
>
> > Details on the empirical evaluations:
>
>  - The likelihoods in the empirical evaluations are computed based on the inversion theorem. For discrete leaves and Gaussian leaves, the likelihoods can be computed analytically. While for $\alpha$-stable leaves, the likelihoods are computed via numerical integration using quadrature. In general, it depends on the form of the characteristic function that is assumed at the leaf nodes. For example, one might relax the assumption that it is specified by a parametric family and could learn the characteristic functions directly. However, doing so is more involved as one has to ensure that the properties listed in Section 3.2. are still fulfilled. We believe this to be a promising future avenue. Once likelihoods at the leaves are computed, they are propagated bottom up following the inversion theorem in Section 4.1. We will improve the description in the updated manuscript.
>
> > Acronyms:
>
>  - We will reduce the use of acronyms (RS, SL, etc.) in the updated manuscript for better readability and thank the reviewer for pointing this out.
>
> > Regarding the question on the relationship between CFD and likelihood:
>
>  - Connections between maximum likelihood estimation (MLE) and minimizing a distance (e.g., the CFD) to the empirical characteristic function (ECF) is indeed an interesting question. Minimizing the CFD to the ECF can be beneficial if no tractable form of the likelihood exists but the characteristic function can be tractably evaluated. As discussed in prior works (e.g., [Yu, 2004]), minimizing a distance function to the ECF is most related to moment-matching approaches, but can result in more accurate fitting results. An interesting future direction could be a hybrid objective in which tractability of either the likelihood function or the characteristic function is exploited. We thank the reviewer and will add further detail and a discussion on the topic to the revised version.
> ***
> [Sasv&#225;ri, 2013] Zolt&#225;n Sasv&#225;ri. "*Multivariate characteristic and correlation functions.*" volume 50. Walter de Gruyter, 2013.
> [Yu, 2004] Jun Yu. "*Empirical characteristic function estimation and its applications.*" Econometric reviews, 2004.

---

> > ### Comment · Reviewer_1MK1 · 2023-08-17
> >
> > Thank you for taking your time to answer the questions. I look forward to reading more about the connections between MLE and minimizing CFD in an updated version of the paper. I'm in favor of accepting this paper.

---

### Decision · Program_Chairs · 2023-09-21

**Decision:**

Accept (oral)

**Comment:**

This paper proposes a great new member of the probabilistic circuit family, representing characteristic polynomials. It might turn out to be a seminal paper in modeling tractable distributions.